# Sexuality in Women with Fibromyalgia Syndrome: A Metasynthesis of Qualitative Studies

**DOI:** 10.3390/healthcare11202762

**Published:** 2023-10-19

**Authors:** José Granero-Molina, María del Mar Jiménez-Lasserrotte, Iria Dobarrio-Sanz, Matías Correa-Casado, Carmen Ramos-Rodríguez, Patricia Romero-Alcalá

**Affiliations:** 1Nursing, Physiotheraphy and Medicine Department, University of Almería, 04120 Almería, Spain; jgranero@ual.es (J.G.-M.); ids135@ual.es (I.D.-S.); mcc249@ual.es (M.C.-C.); 2Faculty of Health Sciences, Universidad Autónoma de Chile, Santiago 7500000, Chile; 3Fibromyalgia Association of Almería, 04006 Almería, Spain; carmenrr@cop.es; 4Faculty of Health Sciences, University of Almería, 04120 Almería, Spain; patriciaromeroalcala@gmail.com

**Keywords:** fibromyalgia syndrome, sexuality, female sexual dysfunction, qualitative research

## Abstract

Fibromyalgia syndrome (FMS) is a nonarticular rheumatic syndrome which presents as chronic musculoskeletal pain, stiffness and body aches. FMS affects approximately 2.5% of the population, mostly women. FMS causes physical and psychological problems and reduces quality of life. The objective of this study is to identify qualitative evidence about experiences of women diagnosed with FMS about their sexuality. Methods: Metasynthesis of qualitative studies. The search included articles published between 2000 and June 2023 on the PubMed, WOS, CINAHL, SCOPUS, and SCIELO databases. Results: 450 articles were found through the initial search, of which, only nine fulfilled the criteria and were included in the thematic synthesis. From this analysis, three main themes emerged: (1) “I want to, but I can’t”: FMS causes a shift in feminine sexuality. (2) Resetting sex life and intimacy. (3) Taking charge of a “new sexuality.” Conclusions: Women with FMS suffer from limitations of their sexuality that affect their partner. Pain, stiffness and a loss of desire make sexual encounters difficult. Becoming aware of this and striving not to lose their sexuality is key to coping with this problem. Women and their sexual partners can change roles and encourage communication, games, foreplay or touching. The use of lubricants, physical exercise and complementary therapies, along with social, professional and partner support, are key to coping with FMS.

## 1. Introduction

Fibromyalgia syndrome (FMS) is characterised by chronic widespread pain, fatigue, and sleep disturbances [1]. FMS is a nonarticular rheumatic syndrome that presents with chronic musculoskeletal pain and multiple points of bodily pain when pressure is applied. Prevalence of FMS in the Euro zone is estimated to be around 2.64% [2], being about 1.5% in France, 2.6% in Germany and 2.4% in Spain [3]. Differences in estimates may be due to the study population, designs, measurements and diagnostic criteria applied [4]. Prevalence rates of FMS are higher in women than in men, ranging from 3.5 to 5.5%, and increase with age [5,6]. FMS alters the physical and emotional health of women [7], and also affects their sexual health as it is associated with fatigue, pain and libido disorders, generating sexual dysfunction [8,9].

SFM is a chronic disease of unknown physiopathological mechanisms, with a predominantly clinical diagnosis. Prevalent symptoms include generalised musculoskeletal pain, hyperalgesia, paresthesia and joint stiffness [10], sleep disorders [11], stress and fatigue [12]. Given the difficulty in establishing a clear diagnosis, the American College of Rheumatology (ACR) established a series of diagnostic criteria such as the presence of musculoskeletal pain lasting more than 3 months, a minimum of 11 tender points to the touch (out of a total of 18), as well as the addition of a symptom intensity index measuring pain, drowsiness and fatigue [13,14]. SFM also affects the mental and emotional well-being of women by decreasing their quality of life [15,16].

FMS has been associated with major depression, mood, bipolar and panic disorders, along with coping difficulties and social stigma [17]. FMS directly affects female sexuality [11], and is associated with fatigue [18], pain [15], lubrication and pelvic floor muscle problems [10], hypoactive sexual desire and self-image disorders [19]. This situation can cause women to have decreased libido, lack of receptivity and sexual avoidance behaviours [20]; with decreased/absence of sexual intercourse and increased risk of relationship breakdown [21]. This situation is compounded by the negative side effects of pharmacological treatment [20], which greatly enhance the problems described above.

While FMS research has focused primarily on physical, psychological problems, treatment, and therapy [22,23], little is known regarding women’s own experiences of their sexuality [9,24]. Qualitative methodologies have proven useful in comprehensive FMS research, but the scientific literature on the effects of FMS on female sexuality needs to be explored. Understanding experiences of women diagnosed with FMS about their sexuality could be important for removing barriers, bringing about improvements, and developing specific protocols for care. Although several studies focus on the experiences of women with FMS [9,24,25], including review studies [26,27,28], a synthesis of aggregate data is needed to gain a deeper understanding of the phenomenon in order to guide clinical practice, raise new hypotheses, and improve quality care for these women. The research question guiding this review is: What are the experiences of women diagnosed with FMS, in terms of their sexuality? The objective of this study is to identify qualitative evidence about the experiences of women diagnosed with FSM regarding their sexuality.

## 2. Materials and Methods

### 2.1. Design

A systematic review of qualitative studies was performed. Metasynthesis involves the inductive analysis, assembly and categorisation of findings on the basis of similarity of meaning, generating a set of statements that bring together and reflect on knowledge about an area of study.

### 2.2. Search Methods

Bibliographic searches were conducted on the PubMed, WOS, CINAHL, SCOPUS, and SCIELO databases for qualitative studies in English and Spanish, published between 2000 and June 2023. The PRISMA 2020 statement, an updated guide for reporting systematic reviews, was used in this study [29]. The SPIDER method was used for qualitative research (Sample, Phenomenon of Interest, Design, Evaluation, Research type) [30]. In order to perform the search, the search terms used were divided into three areas. Firstly, the terms “sexual”, “sexuality”, were joined with the Boolean operator “OR”, this search was performed with “women”, “girl”, “women’s health”, as well as “fibromyalgia”, “fibromyalgia syndrome”, “muscular rheumatism”, “chronic fatigue disorders”, “chronic fatigue syndrome”, and “qualitative research”. After performing these searches separately, they were joined using the Boolean operator “AND” ((sexuality) OR (sexual)) AND ((women) OR (Girl) OR (Women’s Health)) AND ((Fibromyalgia) OR (fibromyalgia syndrome) OR (muscular rheumatism) OR (chronic fatigue disorders) OR (chronic fatigue syndrome)) AND (qualitative research). This search was supplemented by a manual search of grey literature.

### 2.3. Inclusion and Exclusion Criteria

Inclusion criteria: Women over 18 years of age diagnosed with FMS at least 1 year prior. Qualitative research articles (descriptive, phenomenological, ethnographic, grounded theory, etc.), or mixed methodology articles. Original research, with access to full text, published in English or Spanish, between 2000 and 2023. Exclusion criteria: non-primary articles, editorials, abstracts or opinion pieces.

### 2.4. Search Results

A 5 stage selection process was performed: elimination of duplicates, title selection, abstract review, full paper review and reference tracking. A total of 131 studies were identified; however, only 9 articles met the inclusion criteria and were included in this review (Figure 1).

### 2.5. Quality Assessment

Each primary study was assessed using the Joanna Briggs Institute’s Qualitative Assessment Rating Instrument (QARI) [31]. The included articles were considered to be of high quality with respect to objectives, design, analysis and results, providing useful knowledge on the topic (Table 1). No studies were excluded after quality assessment; however, 3 studies had significant methodological limitations.

### 2.6. Data Extraction

Researchers removed all duplicate records, analysed selected studies by extracting data on author(s), year, country, design, philosophical perspective, sample, data collection, data analysis, age and research focus. Two authors (MMJL, PRA) independently performed data extraction, discussed discrepancies, and reached an agreement. Finally, the references of nine included papers were reviewed.

### 2.7. Data Synthesis and Analysis

The included studies were analysed thematically. The synthesis was undertaken by (PRA an MMJL), and verified by (JGM). Two independent reviewers with expertise in FMS and qualitative research verified the results. The thematic synthesis of qualitative data (Table 2) included line-by-line coding, developing descriptive themes and generating themes and sub-themes in three stages [32]:

### 2.8. Rigor

To check the validity of the review, we maintained structured summaries of all original studies, and also checked whether the findings were transferable to other research contexts. Following the thematic synthesis, we examined the studies’ contributions to the final analytical themes and intervention recommendations.

## 3. Results

The nine qualitative studies comprised a total sample of 132 women diagnosed with FMS from Brazil, Colombia, Canada, Spain and the USA, aged between 18 and 61 (Table 3). Most of the studies were conducted in Spain (4). Two studies conducted in Colombia [34,35] and two studies conducted in Spain have the same data collection [9,24], although they focus on different aspects of sexuality. Thematic synthesis is an inductive process in which three themes and 18 sub-themes emerge (Table 4).

### 3.1. “I Want to, but I Can’t”: A Shift in Feminine Sexuality

A common denominator in the experiences of the patients in our synthesis is the presence of debilitating pain that affects their sex life. Although the testimonies differ, this pain is accompanied by stiffness, decreased desire, irritability, altered emotional state, poor mood and decreased quality of life. Although women try to cope, there is a decrease in the frequency and quality of sexual encounters, fluctuating pleasure and fewer orgasms. Taking drug treatments can accentuate the problem, and women with FMS may avoid sexual encounters and have sex to please their partner.

#### 3.1.1. Pain/Stiffness Limit Pleasure and Desire

Women with FMS typically experience pain and stiffness. Pain is accompanied by generalised bodily stiffness at variable periods of time, exacerbated by flare-ups. These symptoms, along with a lack of female vaginal lubrication, directly affect penetration, making it difficult and uncomfortable for the partner during intercourse. 


*The pain is concentrated in the vaginal area, the moment of penetration is really painful for both partners. We didn’t have these problems before, it was all because I was diagnosed with FMS.*
[34]

The pain is often accompanied by generalised stiffness. Women with FMS note a semi-numb, tense feeling, with a paresthetic sensation throughout their bodies. There is muscle stiffness in the legs that interferes with activities of daily life such as walking, but also negatively affects sexual desire. Women report that the pain blocks their emotions, they lose the desire to have sex and it makes orgasm difficult. After initiating intercourse they try to endure the pain, but end up exhausted and have to stop.


*Sometimes you have to say, ‘Stop, stop, … you’re hurting me, I can’t do it’. Or he holds you and … ‘Ow, you’re hurting me!*
[24]

In this situation, changes in position do not always improve the pain; on the contrary, they can increase the discomfort during intercourse. 


*Changing positions during our encounters hurts a lot; I didn’t have this problem before, now it hurts anytime we do anything out of the ordinary…*
[35]

#### 3.1.2. Irritability and Low Mood

The intensity of the pain causes irritability, discomfort and moodiness. Women suffer from headaches and general body discomfort that affects their sexual function. During flare-ups, women with FMS do not want to be touched by anyone, including their partner. They report that it hurts “down to their toes”, and sex is the last thing on their minds.


*When I am in such intense pain, I get a bad temper and tell him to leave me alone, that I do not want to be with anyone, nor with myself. By the pain I get bad and insult him without him deserving it.*
[35]

The generalised stiffness prevents them from relaxing and enjoying sex, and they have severe difficulties in reaching orgasm. Women report that this feeling is difficult to explain, especially to their partners, as reported in a focus group:


*I had a lot of discomfort doing it (coitus), some pain here (vulva) and I didn’t have one (an orgasm). I was very nervous, I couldn’t relax, I wasn’t enjoying it. How can you always explain that? It’s like… it’s a bit ridiculous.*
[24]

It is therefore a “hidden” pain, which affects their state of mind. They do not want to talk about it, they do not know how to explain what is happening to them, they do not want to be asked about it, so they generally keep quiet and say nothing.


*Get up every day and hear that it is hurting me here or there, I know it must be exhausting. It is enough that I must deal with this daily to make another person deal with the same thing … I know it’s difficult and that’s why I always try to show a good face and avoid him knowing that it hurts me.*
[35]

#### 3.1.3. Decreased Frequency/Difficulty Reaching Orgasm

Pleasure and desire are fluctuating; some women maintain their desire for sex, but most do not. They feel that they cannot enjoy it, they find it difficult to reach orgasm and want to finish as soon as possible. This increases their irritability, as their mind is not focused on sex.


*My mood is directly related to the amount of pain I’m in. My irritability is directly related to the amount of pain I’m in… so, if I’m in a bad mood, feeling irritable and in pain, I’m not going to want to have sex.*
[36]

The frequency of sexual intercourse drops from daily to weekly, from weekly to monthly. This is very difficult for the partner to cope with, especially in young male partners.


*Before I had the disease, when I was 35 years old, we were able to have sex once or twice a week. Since I was diagnosed with fibromyalgia about five years ago, the frequency has dropped to about once every two months.*
[39]

#### 3.1.4. Pharmacological Treatment Does Not Help Sexuality

Psychiatric comorbidity does not help to improve the sex life of women with FMS. Many are diagnosed with psychological disorders such as anxiety, depression, etc. They feel misunderstood and stigmatised by professionals. Taking antidepressants, analgesics or opiate derivatives worsens their predisposition towards sexual intercourse.


*Took it, but the muscular weakness was so bad in my legs that I couldn’t get up… I told them that I wouldn’t take Tramadol^®^ anymore or any other drug. They told me, “Well then, next time you can go to mental health.*
[9]

#### 3.1.5. Having Sex “for Your Partner”: Avoiding Encounters

Women with FMS avoid circumstances that could potentially lead to a sexual encounter with their partner. Sometimes they do not want it, sometimes they get angry at the lack of understanding. Excuses vary from drowsiness to headache, and women prepare their partners not to expect something (sex) that is never going to come.


*It’s not that I say I don’t want to, I just do everything in my power to make sure the situation doesn’t arise (…) from the afternoon on, I start telling my husband that I feel really badly and that way I can ensure that nothing sexual is going to happen between us that night.*
[34]

Other times they feel guilty about denying their partner something they feel they deserve and cannot give them due to their situation.


*When my husband initiates sex with me (I never do), the first thing that comes to my mind is that I will have pain in my legs and hips for more than a week because of the movements and postures. This takes away all my urges, even though I might have some desire to make love. If I see that my husband is very eager, then I’ll give him that reward because the poor guy is very good to me and he deserves it.*
[39]

Over time, women show more direct refusal to have sex. “So one day I made the decision to tell him that I just didn’t want to have sex because I didn’t feel good and I didn’t know if that was going to change” [34]. Other times they succumb to their guilt and permit sexual intercourse to satisfy their partner’s needs, they strive to satisfy them and not lose them. This is how one woman puts it:


*I do it for my husband. Yes, it’s for him, because I don’t feel like having sex at all.*
[36]

### 3.2. Resetting Sex Life and Intimacy

Acceptance of the new situation is key to the sexuality of women with FMS. Changes in body self-image, desire, partner role or social behaviour must be addressed. Women tend to feel misunderstanding, stigma and vulnerability towards their condition. Their partner is key in the process of acceptance and change. Fear of abandonment fuels most behaviours.

#### 3.2.1. I Know it’s Not Mutual

FMS leads to loss of libido, therefore the partner feels that the woman is never interested in sex, has changed, or is not like she used to be. Women feel that they cannot meet their partner’s sexual demands, or are not fulfilling as wives, which generates frustration, anguish and silence.


*I feel terrible for not being able to be more affectionate … but I know that this is because of the pain, because of the anguish I feel all the time.*
[35]

Women with FMS feel guilty, and have sex because they see the lack of affection their partner receives as unfair and want to reciprocate. Faced with frustration, they sometimes even consider going their separate ways, knowing that they no longer meet their partners’ expectations.


*He (my partner) knows that I don’t do it because I feel like it, but to satisfy him, obviously. There are times when he finishes (orgasm) and you,…mmm,… you don’t, and he also feels guilty and frustrated.*
[24]

#### 3.2.2. Bearing the Moral Burden

Although it depends on cultural, educational or aspects of a certain country, patriarchal behaviours in the relationship undergo changes. When faced with the vulnerability and fragility of women with FMS, husbands tend to take on the role of carer. 


*He [my husband] takes care of me, I am the wife, so it is right for the man to take care of the woman […] the woman takes care of the man with the housework, washing, ironing, tidying, giving [sexual] pleasure.*
[33]

From a religious perspective, as a partner, women equate a man’s love with sex, as if their bodies belonged to each other. They consider masculine sexual satisfaction as an obligation to be tended to as wives.


*Poor him [of the husband] […] We [the wife] have to understand that he has [sexual] needs.*
[33]

Social isolation is also an issue, as women observe a loss of close friends, work colleagues and the social support they once had, reducing their quality of life. 


*If you’re lucky enough to have someone who’ll stand by you and understands what you’re going through, or has an inkling at least, OK, but if not … each go their separate ways, that’s how it is.*
[38]

#### 3.2.3. Managing Misunderstandings/Support

Women allude to their partner’s lack of understanding of FMS, not understanding what they are going through, not empathising with them and not being able to put themselves in their position. The partners see the situation as unreal: their wives seem older, they suddenly do not feel like having fun or being social, and get cross easily. As one woman states: You lose your sex drive, and then everybody pulls a long face [38]. The feeling of having support from their partner varies; at first the partner may feel uncomfortable, but little by little, they start accepting their fate, and mould their social lives and sexual intimacy to the woman’s current situation.


*I feel like he understands me, that he makes a big effort to understand my pain. At the same time, I try to forget my pain, and put in an effort on my part so we don’t have to stop doing the things he loves.*
[34]

Over time the relationship strengthens, as partners understand each other better, they know what each person expects and when it is best for sexual relations to happen. “*Over time we have learned to support each other mutually; all the things we’ve gone through have made us stronger*” [34]. Some women even do not understand when they receive excessive support from their partners.


*Since I have fibromyalgia, he has become too sensitive; if I cry, he cries with me. That makes me feel accompanied, that I think he understands my pain. Sometimes he gets so bad for my pains, that it’s my turn to console him and tell him that everything will turn out well.*
[35]

Along with a lack of understanding from their partners, women with FMS also note a lack of understanding from professionals. Participants express frustration upon not being understood by medical professionals [37]. Doctors and nurses usually do not know the implications of FMS on patients’ sex lives, or they minimise their concerns and attribute it to the state of their mental health. The sexual implications of FMS are not considered during examinations, as one woman states, “*It’s as if you have to just figure it out on your own*” [9], and these topics are never addressed:


*I left the consultation feeling really down… you feel like they’re not listening to you. He told me that he was going to send me to a psychiatrist. There was no sensitivity… so I wasn’t exactly going to talk or ask about sex.*
[9]

#### 3.2.4. Vulnerability…of My Relationship

Women’s self-image changes with FMS, as they do not feel beautiful, attractive or desired although their partner tells them so. Treatment with steroids makes them gain weight. They feel bigger, and it is a dark time. They do not want to be seen, and want to stay hidden.


*I don’t feel pretty, … I want to hide in the dark.*
[36]

They feel extremely vulnerable, and worry about the repercussions of their situation on their relationship, which also becomes vulnerable. 


*Getting up every day and hearing that it’s hurting here or there, I know it must be draining. It’s already bad enough that I have to deal with it on a daily basis, without having to make someone else deal with it every day too.*
[34]

#### 3.2.5. Faking It for Fear of Abandonment

Women with FMS feel frustration and vulnerability within themselves and in their relationship. Aware of the danger of being abandoned, they start to feign interest in their sex life: “*He asked me if that was ok, if I was having fun, and I told him I was, even though I was actually faking it*” [35]. Profound changes in a couple’s romantic life may give rise to resentment, and partners may become distant or colder, as they expect understanding rather than pity. 


*I wonder to myself if he would ever leave me one day, because it could happen, he might get tired of dealing with it and someone else comes along who wants to go out and have fun, who can do the things he wants to do, who has things in common with him.*
[34]

### 3.3. Taking Charge of a “New Sexuality”

FMS alters women and their partner’s sex lives, and facing these challenges lies in accepting the changes and adapting to a “new sexuality” by developing several different strategies.

#### 3.3.1. Striving for My (Our) Sex Life

Women strive to give priority to their sexuality as a first step in order to not lose it. They understand that sex is not everything, but it is part of the glue that holds the couple together. Even if they cannot perform as they wish, they make an effort.


*For me, yes, obviously, it’s like before, just as important. I throw myself into it, because. I have to do things, I have to have a life, sex too.*
[24]

It is important to feel attractive to their partner, or for other men, feel good about their appearance and feel wanted. Wearing more beautiful clothes and wearing makeup helps them feel more desirable:


*… If I can feel pretty, maybe I can feel better about myself…it will help me feel more desirable, which in turn will hopefully bring the sex back into our lives.*
[36]

#### 3.3.2. Forgetting the Past: Taking the Initiative

Women must learn how to cope with guilt, and forget the past. Increasing communication with their partner may help to recognise where there are problems and solve them.


*I love you, and yes, you have fibromyalgia, but we’ll get through this together, you have to stop feeling guilty for not being able to do it [sex]. I’m fine…*
[36]

Women must take the initiative in their sex life and relationship. It is they who know when they are in pain or not, and when they are able to have sex: “*He’ll wait until I tell him I’m ok and then we’re ready to go*” [36]. Couples must avoid comparisons with the past, look for new information and seek professional help


*It would be good if they explained it to the partners, that when you have a strong chronic pain, you are physically not up to having a sexual relationship, and that if you need to rest while having sex, that’s quite normal, because physically, your body needs to rest.*
[9]

#### 3.3.3. Changing Habits: Finding the Right Moment

The partner must be willing to change their sexual habits, for example, if they are accustomed to having sexual encounters at night and now their wife needs to go to bed because of fatigue, they will have to find another time. As one patient recounts: 


*I have to do it when I’m not tired—it’s not so much the frequency, but of finding other ways of doing it, whereby it’s not painful, not because of having sex, but of the correct position.*
[38]

Communication is paramount, as the partner must comprehend that it is not always the right time to have a sexual encounter. Women with FMS must show signs that “*today is not the day*”, and their partner must withdraw their advances and not pressure the woman, in order to avoid negative feelings.


*We understood that we could not always have relationships, that we could when the pain was bearable.*
[35]

Other women insist it is important to initiate, not ignore attempts by their partner to initiate, regardless of if it is cut short or ends in orgasm. 


*You don’t feel like it until you actually start… then you get into it and you feel like it more.*
[24]

#### 3.3.4. Getting to Know Each Other: Prioritising Play and Touch

Women with FMS recognise the importance of making certain preparations for sex. Putting cushions on the bed, regulating foreplay or not overextending oneself due to the risk of fatigue can help. This is explained by one woman regarding penetration. 


*There is not much movement after penetration either, my husband knows it has to be gentle and quick.*
[39]

Women have moved from coitus-centred, genital sexuality to exploring, playing, caressing and getting to know each other. There are clear improvements in partner satisfaction and in the relationship. The woman feels more comfortable and enjoys her sex life more.


*Sexuality for me before was from the genital, that is, from intercourse; we did not worry about what we felt… Now we explore ourselves, we talk, we laugh, we try things.*
[35]

#### 3.3.5. Changing Positions and Using Lubricants

Women with FMS do not refuse to change positions, but when it hurts or they get cramps, they are unable, so they choose to explore non-painful positions that give them pleasure. There are positions they cannot assume or must change quickly.


*Sometimes, if I’m having a bad day, I may have to say, “Hey, my knee hurts, let’s not have sex in that position, let’s do it this way.”*
[36]

#### 3.3.6. Exercise and Therapy

After being diagnosed with FMS, women usually turn to exercise and therapy as adjuvants to medical treatment. Psychotherapy can help with diagnosis and have a positive impact on pain. Regular physical exercise also seems to have a positive impact on pain and fatigue. One woman adds:


*With physical exercise, yes, because you’re more active, you feel better (laughs). It helps you to get into the mood more.*
[24]

#### 3.3.7. Social and Professional Support

Sexual problems associated with FMS are not on health professionals’ agendas. Little is discussed with doctors, nurses or physiotherapists, who also do not ask questions. This places an additional burden on patients, who are forced to raise the issue in the consultation room for treatment or referral to other specialists.


*Sexual problems in fibromyalgia are never discussed with doctors. They never bring it up, it seems that it doesn’t exist or that we don’t have sexuality. We don’t bring up the problem either.*
[39]

Participants in the various studies emphasise the importance of patients’ organisations or groups as a primary means of support. Within these groups, women share problems with others in the same situation, and find a listening ear and understanding.


*Since I’ve been coming to the association, I’ve come to understand FM. Now, being here with other people and seeing they have the same symptoms as you and everything… it’s like you understand the illness better.*
[9]

## 4. Discussion

The objective of this study was to identify qualitative evidence about the experiences of women diagnosed with FSM regarding their sexuality. According to our results, chronic pain limits satisfactory sexual relations and significantly influences the lives of women with FMS [35]. Women have a lower pain threshold and greater sensitivity to pressure and temperature [40]. Several studies agree that pain inhibits sexual desire and satisfaction compared to healthy women, and sexual dysfunction is essentially related to the severity of coital pain [41]. Some meta-analyses agree on the association between female sexual dysfunction and FMS [42], and although they recommend changes in position during sexual relations, our review shows that this may produce pain and is not always possible [34,35]. 

Coinciding with our findings, pain, stiffness, fatigue and lack of sleep especially contribute to the loss of sexual health in women with FMS [43]. Some studies report that the symptoms may even lead women to stop the sexual activity once it has started [24]. Pain appears to be associated with low mood and anxiety in women with FMS [44], and depression is common, related to problems with desire and arousal [19,45]. Our metasynthesis confirms that women suffering from FMS pain do not want to be with anyone [35], rejecting any contact with a partner. They report that pain negatively influences the frequency of sexual intercourse and their ability to have orgasms [19,20,45]. It is a pain they define as “hidden”, and they learn to live with it, because the treatment options for FMS are limited and focused solely on symptom relief [46]. According to our review [36], women can also suffer from irritability, which leads them to disregard sex. Much controversy remains regarding diagnosis, treatment and assessment [47]. As in other chronic fatigue syndromes [48], women with FMS should be evaluated and treated for psychiatric conditions. Clinicians should consider how treatments could be tailored to individual symptoms, weighing the benefits and acceptability, when prescribing medications to patients with fibromyalgia [49]. Our results confirm the negative repercussions of opioid drugs such as Tramadol on women’s physical fitness and sexual activity [9]. Participants have said that assessing the impact of such treatments on sexual activity is a critical issue [9].

Chronic pain and low moods push women towards sexual avoidance behaviours with their partner, and this emotional distance makes closeness and the ability to negotiate other forms of sexuality nearly impossible. If women with FMS perceive their partners as unsupportive or distant, they are less likely to communicate their needs or share intimate spaces. This is reaffirmed by the studies in our review [34,35], which state that women with FMS perceive sexual encounters as a gendered obligation. Depressive symptoms and antidepressant medication are associated with decreased sexual desire in these women [50], which also could be related to avoidance behaviours. The results of our study indicate that women experience contradictory feelings, forcing themselves to engage in sex with their partners, while doing whatever they can to avoid these encounters. In the case of religious women, some feel an obligation to satisfy the sexual needs of their male partner, as their wife [33], which creates a situation of distress. Over time, avoidance behaviours develop into overt rejection and direct communication to the partner of their refusal of sexual relations [36].

Our results indicate that women are particularly concerned about comorbidities that negatively affect their sexuality. Women with FMS are three times more likely to report pelvic floor dysfunction, sometimes accompanied by urinary incontinence, mixed incontinence and flatulence [51]. FMS can also affect psychological and physiological processes in women with overactive bladder, [52]. Women have told us that their partners do not understand this situation, and know that they have sex with them only to please them [24], which leads to lack of communication, feelings of guilt and the progressive abandonment of their sex life. Self-image problems and comorbidities increase feelings of vulnerability of these women, leading to an increase in doctor visits [53]. As our metasynthesis indicates [24], women do not feel beautiful, attractive or desirable, rather, they feel that their bodies have changed, they look fat, their skin condition is worse, and they do not feel like putting on make-up or dressing up to feel more attractive.

Evidence points to the need for support and an interdisciplinary approach to sexuality in women with FMS. As our review points out, social support is key in coping [9]. Women and their partners must recognise the situation and adapt to a new type of sexuality, both taking the initiative [24]. Women are committed to forgetting their past sex life, as they are under different circumstances and currently have a new kind of sex life. It is necessary to find opportune moments for sexual intercourse; the woman must give signals and the partner must adapt [35,38]. For women suffering with FMS, owning the problem and forgiving themselves is important [54], and improving their self-image to feel attractive can also be beneficial [36]. Our findings suggest that not refusing sexual activity from the outset may be an option, even if the activity eventually has to stop because every woman and every relationship is different [24]. Communication between partners and commitment to changing their habits is essential [35,38]. Emphasis on play, touch and preparation is key for women [39], along with lubricants and exploring less painful positions [36]. Physical exercise and complementary therapies can help [24], but women must be open to learning about them and work with qualified professionals. Although specific improvements have been reported, there is little evidence regarding the influence of these on sexuality [55,56,57]. Professional support and training, a good doctor–patient relationship and continuity of care are essential [39]. However, our results indicate that many women have not perceived such improvements and have lost confidence in professionals, turning instead to each other within patient support groups [9]. FMS is also a challenge for professionals such as physiotherapists, but those specialised in treating pelvic floor problems may contribute to improving sexual problems [58]. Social support is fundamental, and patient, family and partner associations are key elements [9].

**Limitations:** The credibility of metasynthesis lies in finding, extracting and analysing qualitative data from studies using a systematic method of revision. Transferability may be limited because women with FMS have disparate cultural and religious backgrounds, limiting results to more specific areas. Varied definitions and the broad nature of human sexuality could lead to greater heterogeneity of findings. The authors may have influenced the process of data extraction and synthesis, but they are experienced qualitative researchers. This study combines the limitations inherent to qualitative studies along with the limitations characteristic of literature reviews. Therefore, the results have a limited capacity for extrapolation.

## 5. Conclusions

Women with FMS are aware of the limitations that their clinical condition imposes on their sexuality and that of their partner. Pain, stiffness or irritability translate into loss of desire, difficulty in having orgasms and fewer sexual encounters. The multimorbidity of FMS implies treatments with negative consequences on their sex life. Women do not have a fulfilling sex life, and some focus on satisfying their partner for fear of abandonment. Awareness and coping with the situation means making an effort not to lose their sexuality. FMS resets women’s sex lives, as they know they do not respond to their partner’s demands and feel a moral burden. Women feel vulnerable and guilty. Female FMS patients and their partner must take charge of their sexuality in their new situation. Changing roles and habits during sexual encounters is the responsibility of both the woman and her partner. Games, preparations, touching, lubricants, exercises and complementary therapies, along with social, professional and partner support, are considered key aspects in the process of adapting to a new type of sexuality.

## Figures and Tables

**Figure 1 healthcare-11-02762-f001:**
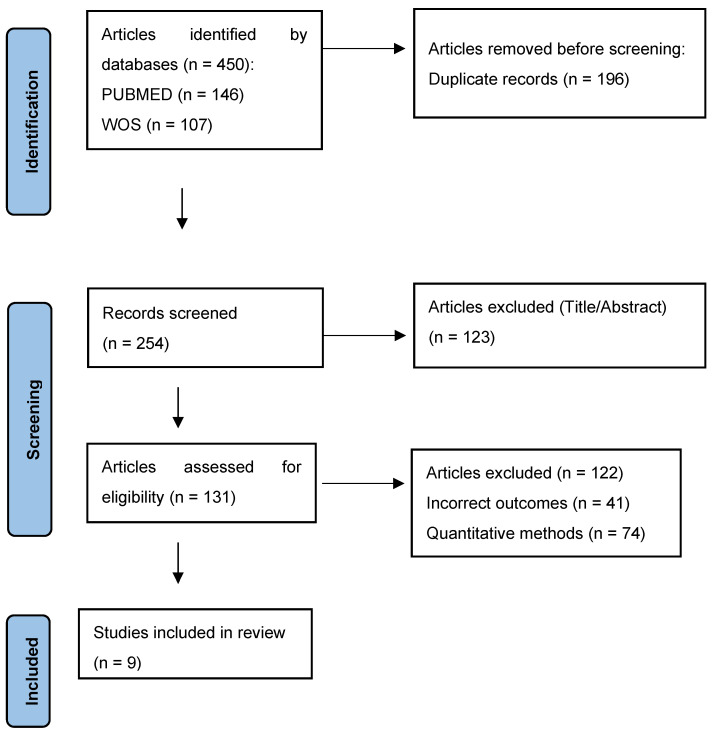
Flow chart.

**Table 1 healthcare-11-02762-t001:** Quality assessment of studies [22].

Article	1	2	3	4	5	6	7	8	9	10
Centurión N.B., et al., 2020 [32]	✔	✔	✔	✔	✔	↔	↔	✔	✔	✔
Sanabria, J. P., and Gers, M. (2019, a) [33]	↔	✔	✔	↔	✔	✔	✔	✔	✔	✔
Sanabria, J. P., and Gers, M. (2019, b) [34]	↔	✔	✔	↔	✔	↔	↔	✔	✔	✔
Santos-Iglesias, P., et al. (2022) [35]	✔	✔	✔	✔	✔	✔	✔	✔	✔	✔
Matarin Jiménez et al., 2017 [24]	✔	✔	✔	✔	✔	✔	✔	✔	✔	✔
Arnold, L.M., et al., 2008 [36]	↔	✔	✔	✔	✔	✔	✔	✔	✔	✔
Briones-Vozmediano et al., 2016 [37]	✔	↔	✔	✔	✔	✔	✔	✔	✔	✔
García Campayo, J. et al., 2004 [38]	↔	✔	✔	↔	✔	↔	↔	✔	↔	✔
Granero-Molina, J., et al., 2018 [9]	✔	✔	✔	✔	✔	✔	✔	✔	✔	✔

✔ Yes, ↔ Unclear. 1. Congruence of philosophical perspective/methodology 2. Congruence of methodology/objectives 3. Congruence of methodology/data collection 4. Congruence of methodology/data analysis 5. Congruence of methodology/interpretation of results 6. Cultural and theoretical context of the researcher. 7. Influence of the researcher on the research 8. Participants represented 9. Research Ethics Committee Approval 10. Conclusions from data analysis/interpretation.

**Table 2 healthcare-11-02762-t002:** Stages in the thematic synthesis process.

Stage	Description	Steps
STAGE 1	Text coding	Recall review question Read/re-read findings of the studies Line-by-line inductive coding Review of codes in relation to the text
STAGE 2	Development of descriptive themes	Search for similarities/differences between codes Inductive generation of new codes Write preliminary and final report
STAGE 3	Development of analytical themes	Inductive analysis of sub-themes Individual/independent analysis Pooling and group review

**Table 3 healthcare-11-02762-t003:** Characteristics of selected studies.

Author Año	País	Muestra (FMSW)	Edad (Años)	Tiempo Entrevista	Data Collection	Data Analysis	Main Theme
Centurión, N.B., et al. 2020 [32]	Brasil	6	40–60	1 h 30 min	DGs	Content analysis	Religion and morals affect women with FMS
Sanabria, J.P., et al. (2019, a) [33]	Colombia	15	23–60	No	IDI	Organisation, segmentation and correlation	Carer roles and gender influence couple dynamics
Sanabria, J.P., et al. (2019, b) [34]	Colombia	15	23–60	No	IDI	Organisation, segmentation and correlation	Feminine viewpoint of FMS influences their erotic expression
Santos-Iglesias, P., et al. (2022) [35]	Canada	16	≥21	60–90 min	SSI	Inductive thematic analysis	Multi-dimensional nature of sexual wellbeing in women with FMS
Matarín Jiménez, T., et al. (2017) [24]	España	13	22–56	40 min	FG, IDI	Gadamer’s phenomenological analysis	FMS affects identity and relationship with partners
Arnold, L.M., et al. (2008) [36]	USA	48	>18	2 h	FG	Strauss and Corbin’s techniques	FMS has a negative impact on quality of life
Briones-Vozmediano, E., et al. (2016) [37]	España	13	24–61	60–90 min	SSI	Thematic analysis	Healthcare providers can help to improve lifestyle in women with FMS.
García Campayo, J., et al. 2004 [38]	España	27	No	60–90 min	SSI, FG	Thematic analysis	FMS limits feminine sexuality, but is not discussed with doctors
Granero-Molina, J., et al. 2018 [9]	España	13	22–56	40 min	FG, IDI	Gadamer’s phenomenological analysis	Lack of formal support regarding fibromyalgia patient’s sexuality

FMSWs = Fibromyalgia Syndrome Womens. IDI = In Depth Interview. FGs = Focus Group. DGs = Discussion Group. SSI = Semi Structured Interview.

**Table 4 healthcare-11-02762-t004:** Themes and sub-themes.

Themes	Sub-Themes
3.1. “I Want to, but I Can’t”: A Shift in Feminine Sexuality	3.1.1. Pain/Stiffness Limits Pleasure and Desire 3.1.2. Irritability and Low Mood 3.1.3. Decreased Frequency/Difficulty in Having Orgasm 3.1.4. Pharmacological Treatment Does Not Help Sexuality 3.1.5. Having Sex “for Your Partner”: Avoiding Sexual Encounters
3.2. Resetting Sex Life and Intimacy	3.2.1. I know It’s Not Mutual 3.2.2. Bearing the Moral Burden 3.2.3. Managing Misunderstanding/Support 3.2.4. Vulnerability … of My Relationship 3.2.5. Faking It for Fear of Abandonment
3.3. Taking Charge of a “New Sexuality”	3.3.1. Striving for My (our) Sex Life 3.3.2. Forgetting the Past: Taking Initiative 3.3.3. Changing Habits: Finding the Moment 3.3.4. Getting to Know Each Other: Prioritising Play and Touch 3.3.5. Changing Positions and Using Lubricants 3.3.6 Exercise and Therapy 3.3.7. Social and Professional Support

SRH = Sexual and Reproductive Health. IMW = Irregular Migrant Women.

## Data Availability

The datasets generated during and/or analysed during the current study are available from the corresponding author on reasonable request.

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
