# Peer review of "Sexuality in Women with Fibromyalgia Syndrome: A Metasynthesis of Qualitative Studies"

_healthcare, 2023, doi:10.3390/healthcare11202762_

Round 1
Reviewer 1 Report
In "Exploring the Impact of Fibromyalgia Syndrome on Women's Sexuality: A Metasynthesis of Qualitative Studies," the authors aim to shed light on the often-overlooked issue of how fibromyalgia syndrome (FMS) affects the sexual experiences and relationships of women diagnosed with this condition.
I think that the paper makes a good contribution to the field by addressing an underexplored area of research.
The paper is well-organized and comprehensively described. The introduction effectively sets the stage for the study by providing essential background information on FMS and its prevalence among the population, mainly women. The objective of the study is clearly stated, and the methods section provides a detailed account of the search strategy, inclusion criteria, and data analysis process. However, minor revisions could enhance the clarity of the methodology section. Specifically,
- Clarify the rationale for the selection of the specific search terms, particularly how they relate to the research topic.
- Specify the criteria for defining "full-length original research articles" to eliminate any potential ambiguity.
- Mention if there were any deviations from the inclusion/exclusion criteria during the selection process, and how these deviations were handled.
In the "Search Results" subsection, the 5-stage selection process is effectively described. To enhance reader comprehension, consider the following suggestions:
- Briefly elaborate on the significance of the 131 initial studies and why they were narrowed down to 9 articles.
- Clarify the criteria used in the "full paper assessment" stage, which resulted in the final selection of the 9 articles.
Author Response
See attached file (Resp Reviewer-1)

Reviewer 2 Report
Dear authors,
It was a pleasure for me to review this manuscript that deals with the sexuality of women suffering from fibromyalgia. I found it a very interesting topic.
With the sole objective of improving the quality of this manuscript, I will allow myself to make a comment:
The discussion as currently written can be considered the results section, but more developed, it is hardly compared with other studies.
It is relevant that the results of this review be contrasted more clearly with the bibliography that has been handled in the study and thus analyze the coincidences or discrepancies if any.
All the best.
Author Response
See attached file (Resp Reviewer-2)

Reviewer 3 Report
Thank you very much for allowing me to review the manuscript titled: “Sexuality in women with fibromyalgia syndrome: A metasynthesis of qualitative studies”. Fibromyalgia is a disease of special relevance in women and deserves special attention. However, there are certain aspects that could be improved:
Introduction:
- Was the PICOs question format followed as a research question?
Methods:
- Have PRISMA criteria been used? The most appropriate thing would be to use the PRISMA flowchart in its latest version.
- Data extraction must be carried out by two researchers independently. Could you confirm if this was the case?
- In table 3 there is disparity when writing the names of the authors of the articles. Please check that the edition is the same in all rows.
Discussion:
- The objective must be included at the beginning of the discussion.
- The limitations should include that this study combines the limitations inherent to qualitative studies along with the limitations characteristic of literature reviews. Therefore, the results have a limited capacity to be extrapolated.
Author Response
See attached file (Resp Reviewer-3)

Round 2
Reviewer 3 Report
Thank you very much for allowing me to review the manuscript once again. The authors have done a commendable job and have addressed all my doubts and concerns regarding the manuscript.